# A high-quality human reference panel reveals the complexity and distribution of genomic structural variants

Jayne Y. Hehir-Kwa[1,*], Tobias Marschall[2,3,*], Wigard P. Kloosterman[4,*], Laurent C. Francioli[4,5,6,7], Jasmijn A. Baaijens[8], Louis J. Dijkstra[8], Abdel Abdellaoui[9], Vyacheslav Koval[10], Djie Tjwan Thung[1], René Wardenaar[11,12], Ivo Renkens[4], Bradley P. Coe[13], Patrick Deelen[14], Joep de Ligt[4], Eric-Wubbo Lameijer[15], Freerk van Dijk[14,16], Fereydoun Hormozdiari[13], The Genome of the Netherlands Consortium[†], André G. Uitterlinden[10,17], Cornelia M. van Duijn[17], Evan E. Eichler[13], Paul I.W. de Bakker[4,18], Morris A. Swertz[14,16], Cisca Wijmenga[14], Gert-Jan B. van Ommen[15], P. Eline Slagboom[19], Dorret I. Boomsma[9], Alexander Schönhuth[8], Kai Ye[20,21,22], Victor Guryev[11]

Structural variation (SV) represents a major source of differences between individual human genomes and has been linked to disease phenotypes. However, the majority of studies provide neither a global view of the full spectrum of these variants nor integrate them into reference panels of genetic variation. Here, we analyse whole genome sequencing data of 769 individuals from 250 Dutch families, and provide a haplotype-resolved map of 1.9 million genome variants across 9 different variant classes, including novel forms of complex indels, and retrotransposition-mediated insertions of mobile elements and processed RNAs. A large proportion are previously under reported variants sized between 21 and 100 bp. We detect 4 megabases of novel sequence, encoding 11 new transcripts. Finally, we show 191 known, trait-associated SNPs to be in strong linkage disequilibrium with SVs and demonstrate that our panel facilitates accurate imputation of SVs in unrelated individuals.

---

[1] Department of Human Genetics, Donders Institute, Radboud University Medical Center, Nijmegen 6525GA, The Netherlands. [2] Center for Bioinformatics, Saarland University, Saarbrücken 66123, Germany. [3] Max Plank Institute for Informatics, Saarbrücken 66123, Germany. [4] Department of Genetics, Center for Molecular Medicine, University Medical Center Utrecht, Utrecht 3584 CG, The Netherlands. [5] Analytic and Translational Genetics Unit, Massachusetts General Hospital, Boston, Massachusetts 02114, USA. [6] The Broad Institute, Cambridge, Massachusetts 02142, USA. [7] Program in Medical and Population Genetics, Broad Institute of Harvard and Massachusetts Institute of Technology, Cambridge, Massachusetts 02142, USA. [8] Life Sciences Group, Centrum Wiskunde & Informatica, Amsterdam 1098XG, The Netherlands. [9] Department of Biological Psychology, Vrije Universiteit Amsterdam, Amsterdam 1081BT, The Netherlands. [10] Department of Internal Medicine, Erasmus Medical Center, Rotterdam 3000CA, The Netherlands. [11] European Research Institute for the Biology of Ageing, University of Groningen, University Medical Center Groningen, Groningen 9713AD, The Netherlands. [12] Groningen Bioinformatics Centre, University of Groningen, Groningen 9747AG, The Netherlands. [13] Department of Genome Sciences and Howard Hughes Medical Institute, University of Washington, Seattle 98105, USA. [14] University of Groningen, University Medical Center Groningen, Department of Genetics, Groningen 9700RB, The Netherlands. [15] Department of Human Genetics, Leiden University Medical Center, Leiden 2300RC, The Netherlands. [16] Genomics Coordination Center, University of Groningen, University Medical Center Groningen, Groningen 9700RB, The Netherlands. [17] Department of Epidemiology, Erasmus Medical Center, Rotterdam 3000CA, The Netherlands. [18] Department of Epidemiology, Julius Center for Health Sciences and Primary Care, University Medical Center Utrecht, Utrecht 3584CG, The Netherlands. [19] Department of Medical Statistics and Bioinformatics, Leiden University Medical Center, Leiden 2300RC, The Netherlands. [20] The Genome Institute, Washington University, St Louis, Missouri 63108, USA. [21] School of Electronic and Information Engineering, Xi'an Jiaotong University, Xi'an 710049, China. [22] The First Affiliated Hospital, Xi'an Jiaotong University, Xi'an 710061, China. * These authors contributed equally to this work.. Correspondence and requests for materials should be addressed to A.S. (email: a.schoenhuth@cwi.nl) or to K.Y. (email: kaiye@xjtu.edu.cn) or to V.G. (email: v.guryev@umcg.nl). [†] A full list of consortium members appears at the end of the paper.

Comprehensive catalogues of genetic variation are fundamental building blocks in studies of population and demographic history, variant formation and genotype-phenotype association. To obtain insights in ancestry and linkage disequilibrium of polymorphic sites it is imperative that such catalogues are haplotype-resolved (phased). Crucial improvements in accuracy and power can be achieved through population-specific panels[1,2]. However, current reference panels contain single nucleotide polymorphisms (SNPs), insertions and deletions of up to 20 bp in length (indels) but only a very limited number of structural variants (SVs) larger than 50 bp in size[3,4]. There is ample evidence that SV's play a major role in evolution and disease[5–9]. Therefore, despite posing substantial technical and methodological challenges with respect to discovery, genotyping and phasing of such SV variants, the integration of SVs into reference panels is crucial for a broad spectrum of studies[10,11].

Recently several population-scale sequencing projects have been undertaken aimed at capturing global genetic diversity[12–15]. In addition, a number of projects have focused on single populations attempting to capture the genetic variability of sociologically and/or historically coherent groups of people for specific variant types[15–18]. For example the UK10K project, which aims at capturing rare variants, comprising SNPs, indels and large deletions used ∼7× whole-genome and ∼80× whole-exome sequencing of nearly 10,000 individuals[17]. A similar subset of variant types were included in the Malay and the Danish genome sequencing projects which both used high coverage (30×−50×), focusing on rare variants that characterize the population[18], de novo variants and the assembly of novel sequence[16].

One of the primary goals of the Genome of the Netherlands (GoNL) project[1,19] was to characterize whole-genome variation in the Dutch population. Our initial reports focused on a whole-genome catalogue of SNVs, small insertions/deletions, and unphased SV deletion events[20,21]. Here, we focus on discovery, genotyping and phasing the full spectrum of structural variants to generate a high-quality SV-integrated, haplotype-resolved reference panel by exploiting two key features of the GoNL project design. First, sufficient coverage (14.5× median base coverage, 38.4× median physical coverage) allows for enhanced genotyping including SVs, as was recently described[18,22,23]. Second, the 769 GoNL individuals originate from parent-offspring families (231 trios and 19 families in which twin pairs are included in the offspring generation), yielding family-based high-quality haplotypes across substantially longer ranges in comparison to statistically phased unrelated individuals[24,25]. In addition to create a haplotype resolved panel, we report several currently under reported variant types, such as deletions 21–100 bp in size, complex indels, inversions, mobile element insertions (MEIs), large replacements and insertions of new genomic sequence[26].

## Results

**Detection of structural variation.** We analysed Illumina whole genome sequencing data derived from 250 parent-offspring families (769 individuals) from the Dutch population to detect structural variants and indels (non-SNVs) using 12 different variant detection tools representing 4 algorithmic approaches (gapped alignment and split-read mapping, discordant read pair, read depth and de novo genome assembly), Fig. 1a, Supplementary Table 1. The results from the different detection tools were combined into a consensus set containing 9 different forms of SVs and indels (simple indels, complex indels, deletions, duplications, inversions, MEIs, interchromosomal breakpoints, novel segments and large replacements). Compared with multiple

public data resources[1,14,26–28], 13.6% of all (simple and complex) indels and 38% of SVs we report are novel (Table 1). To show the specificity of our structural variant predictions, we selected a representative set of candidates across all 9 variant types and performed an independent experimental validation using PCR-amplification across the variant breakpoints followed by Sanger or Illumina MiSeq sequencing (Supplementary Data 1). This yielded a confirmation rate for each variant class of between 80 and 98.6% with the exception of inversions (64.5%), which failed to produce a PCR product in 35.5% of the cases (Table 1). The high rate of PCR failures for inversions might either be a result of false positives, or stem from poor performance of PCR assays, given the frequent occurrence of repetitive and complex sequences at inversion breakpoints.

**Deletions and insertions.** We first focused our analysis on deletions and insertions of DNA sequence relative to the reference assembly. This revealed 646,011 short insertions (1–20 bp), 1,093,289 short deletions (1–20 bp), 24,167 mid-sized deletions (21–100 bp) and 19,840 larger deletions (101 bp–1,467 kb) of which the majority (99,8%) could be genotyped (Table 1, Fig. 1b). We observed an increased number of deletions with size ranges corresponding to SINE and LINE retrotransposition events (Fig. 2, Supplementary Fig. 1). A substantial fraction of the simple indels (11.5%), mid-sized deletions (21.6%) and 41.9% of larger deletions were novel. Of the previously known mid-sized deletions, 79.2% were present solely in our previous GoNL release and in no other call set, emphasizing that this size class has been previously under-investigated.

The consensus set of deletion events were found to be significantly depleted in exonic regions ($P < 10^{-4}$) and UTRs ($P < 10^{-4}$), as well as known disease terms from Online Mendelian Inheritance in Man (OMIM) database, and deletions predicted to result in a loss of function when compared with 10,000 random sets of size matched variants ($P < 10^{-4}$). Further analyses showed that 11 deletions were in transmission disequilibrium (Supplementary Table 2).

**Duplications.** We identified 1,738 tandem duplications, 34.6% of which could be successfully genotyped. This low percentage is likely due to the limitations of current computational methods. The majority of the events were novel (84%, $n = 1,458$) and contained repeat elements such as simple repeats ($n = 914$) or segmental duplication ($n = 194$). A minority of the duplication events ($n = 88$) overlapped a RefSeq gene of which 71 affected at least one exon within a gene and 41 events overlapped at least one exon of a gene with an OMIM disease entry (including susceptibility loci and recessive disease genes) (Table 1, Supplementary Fig. 2).

**Complex structural variation.** A significant proportion of structural variants cannot be described as simple events. Our data show that a sizeable fraction of indels (3%, $n = 52,913$) represent cases where one multibase segment of DNA (2–10 bp) is replaced by another sequence of different lengths (1–11 bp), of which only a minority (17.2%) has previously been described. Furthermore, by combining calls from discordant pair analysis with de novo genome assembly we were able to detect 84 inversions and 60 interchromosomal events, of which 69 and 46 could be successfully genotyped. Interestingly, most of these variants were common with average allele frequencies of 22.8% and 32.2%, respectively. Manual curation of interchromosomal events showed that the majority possessed a polyA stretch at the interchromosomal breakpoints and therefore was likely to originate from retrotransposition events. This observation was

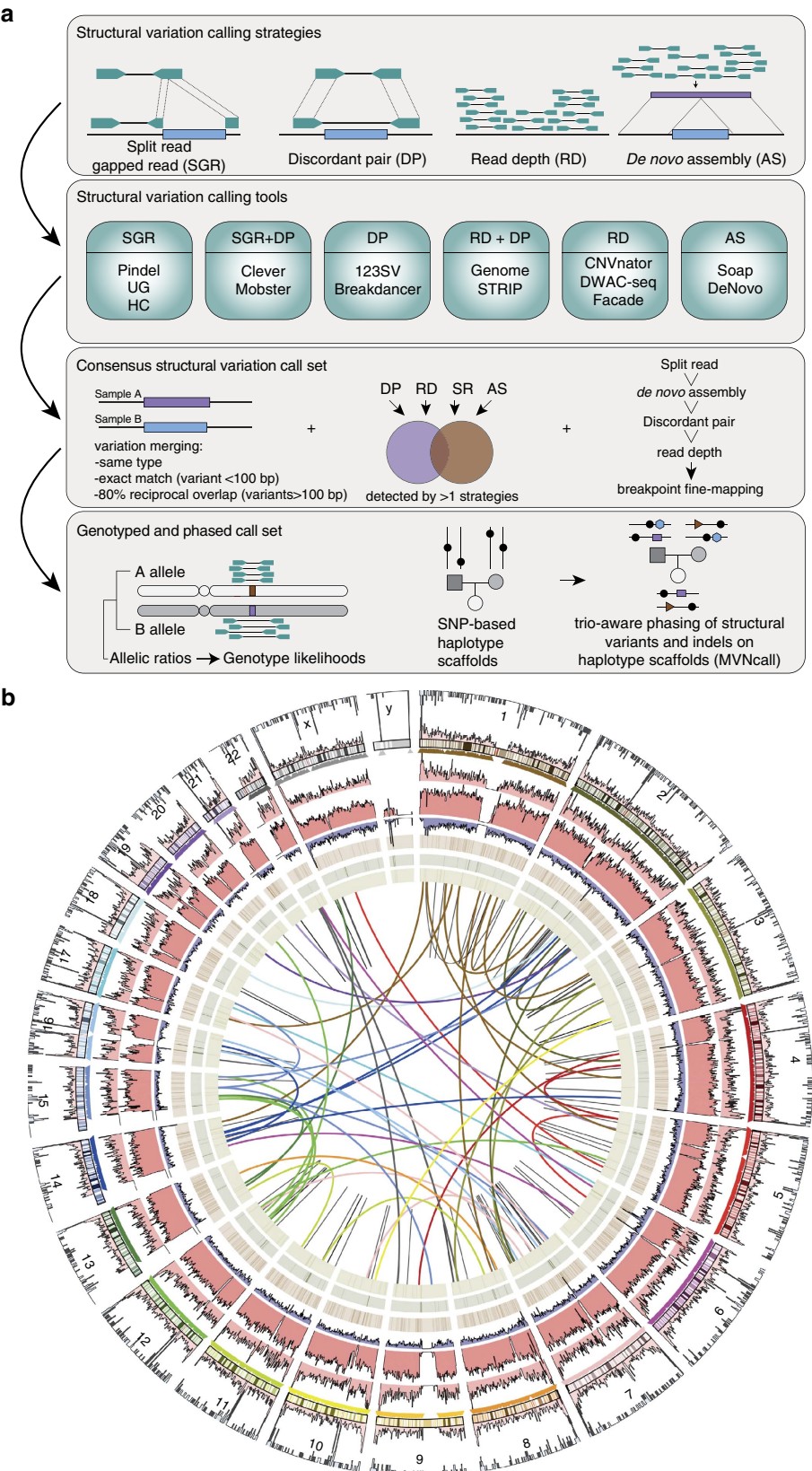

**Figure 1 | Overviews of discovery approach and variant set.** (**a**) Overview of methods used for SV detection, genotyping and phasing within the GoNL project. (**b**) Structural variation consensus set, consisting of large duplications (outer ring), deletions larger than 100 bp (light red), chromosomes, insertions (triangles), mid-sized deletions (21–100 bp), small deletions (less than 20 bp) (dark red) and complex indels (purple). Heatmaps display the insertions of *Alu*, L1 and SVA elements. Inversions are indicated by black arcs in the centre of the plot, and interchromosomal break points (colored based on the source chromosome).

supported by our orthogonal validations, which showed that 11 of the interchromosomal events contained processed parts of known transcripts and were further characterized as gene retrocopy insertion polymorphisms[29] (GRIP) (Supplementary Data 2).

**Mobile element insertions**. MEIs are a common type of retrotransposition-mediated insertions. In total we identified 13,469 MEIs, making it a frequent form of structural variation (23% of SVs larger than 20 bp). The majority of MEIs could be genotyped (99.7%, $n = 13,430$) and were novel (56%) in comparison to those previously reported (Table 1). Non-reference insertions of *Alu* ($n = 8,670$) were the most common form of event followed by L1 ($n = 4,011$), SVA ($n = 781$) and HERV ($n = 7$) insertions. The majority of MEI elements ($n = 8,136$) were located in intergenic regions (Supplementary Fig. 2); however 49 events were predicted to occur within exonic regions, including validated *Alu*Ya4/5 insertions into coding sequences of *OPRM1*, *METTL4* and *ARHGAP28*, as well as a heterozygous *Alu*Yk12 insertion into the last exon of the *EYS* gene, a gene which is involved in autosomal recessive retinitis pigmentosa[30] (Supplementary Fig. 3). The insertion in *ARHGAP28* was observed in three families, while each of the other three coding MEIs were family-specific events. This suggests that these MEIs in the coding part of the genome are relatively recent and/or deleterious.

**Novel segments**. We performed joint *de novo* genome assembly by pooling unmapped and discordantly mapped sequence reads from each family to search genomic segments absent in the genome reference (GRCh37) (ref. 31). Mapping of the resulting contigs to the reference genome allowed us to confirm breakpoints of simple structural variants discovered by alternative approaches. Some of these alignments are consistent with more complex variation types, such as large segmental replacements (Fig. 3, Supplementary Data 3). Contigs that did not match the genome reference partially or completely were analysed separately. The size of unmatched sequences ranges from 150 bp to 133 kb (N50 = 5.6 kb) spanning 22.2 Mb of assembled sequence. A large proportion of these sequences (14.4 Mb) showed discordance between libraries derived from the same individual. Homology searches against a non-redundant NCBI sequence database showed that these segments most likely represent genomic contaminations (Supplementary Table 3). The remaining 7.8 Mb of sequence (11,350 segments) contained sequence not represented in genome reference GRCh37. While the improved GRCh38 assembly places many segments onto the genome map, 4.3 Mb of assembled sequence is still unaccounted (Supplementary Table 3). These segments represented in part difficult to assemble repetitive sequence, but also segments thus far uniquely observed in the Dutch population. Interestingly, while not matching GRCh38, 11 segments match UniGene sequences, and include examples of expressed and potentially functional genes. For example, we identified a novel zinc-finger (ZNF) gene, harboured within an insertion on chromosome 19 (Fig. 4). Although this novel ZNF gene is absent in the human reference (both versions GRCh37 and GRCh38), it has close homology to DNA segments of recently assembled genomes of non-human primates. Mapping of RNA-seq reads from public expression data to a modified human reference genome containing the novel segment showed that the inserted segment codes for a novel spliced ZNF transcript (Fig. 4).

**Load and distribution of structural variants**. Cluster analysis was performed to identify genomic hotspots of structural variants. We confirmed 46 of the 50 deletion hotspots previously reported[32]. Furthermore, when additionally considering duplications, mobile element insertions and inversions 13 variant hotspots were identified, of which 4 have previously not been described[32]. Overall each haplotype within the cohort had on average 758 kb of sequence affected by simple and complex indels and 4.0 Mb by structural variants, amounting to an average of 4.8 Mb of sequence affected by non-SNP variants. On average, every individual carried 436 kb of homozygous simple and complex indels and by 2.4 Mb of homozygous structural variants (Table 1, Supplementary Fig. 4).

**Rare variants**. The majority of small deletions were rare (with minor allele frequency, MAF < 1%, 50.4% of deletions up to 20 bp). Small insertions and mid-sized deletions larger than 20 bp displayed a higher allele frequency (MAF < 1%, 39.7% of insertions up to 20 bp, and 33.5% of deletions longer than 20 bp) (Fig. 2, Supplementary Fig. 5). We stratified each deletion, MEI, short deletion, short insertion and complex indel based on allele frequencies into quartiles. Significantly more exonic events were observed in the first quartile for all variant types tested (Supplementary Data 4). We observed a significant difference in the distribution of indel events occurring within an OMIM gene. More specifically, exonic events affecting OMIM disease genes were more often observed in the first quartile (MAF < 0.325%), as were exonic events involved in a pathway annotated in Kyoto Encyclopaedia of Genes and Genomes (KEGG) database, and those in genes when knocked out in mouse resulting in a phenotype. We observed that rare variants exhibit an excess in deletions larger than 1 kb in size and *Alu*Y insertions. In contrast, deletions that have large overlap with a SINE/LINE repeat occur more frequently in the common events quartile (MAF > 42.5%). This could also be due to rare mobile element insertions where the inserted allele has been incorporated in the reference genome assembly (Supplementary Data 5). These observations may indicate recent alteration in mutational processes, as well as differences due to negative selection against large deleterious variants.

**Effect of structural variants on gene expression**. We obtained gene expression data (based on RNA sequencing data generated from a subset of 115 individuals from the cohort) and tested the effect of structural variants on gene expression. The effects of indels, deletions and duplications on gene expression have been previously described[33]. We explored effects of inversions and mobile element insertions on gene expression. Out of 10 inversions and 139 MEIs that overlap exons or core promoters we found two MEIs with a significant effect on gene expression (Fig. 5). An *Alu*Ya5 insertion was identified in the promoter of the *LCLAT1* gene (Fig. 5a). Samples, which are homozygous for the *Alu*Ya5 insertion display a significantly reduced expression of the *LCLAT1* gene ($P = 6.87 \times 10^{-9}$) (Fig. 5b). We also identified an *Alu*Yb8 element in the last exon of the *ZNF880* gene. This was associated with differential expression of the last two exons of *ZNF880*, resulting from alternative splicing possibly due to effects of the *Alu* element on RNA secondary structure (Fig. 5c). These findings show that some of the less studied types of SV, such as MEIs, can impact gene expression both quantitatively and qualitatively.

**Phasing**. We phased all successfully genotyped simple and complex indels ($n = 1,792,213$) and SVs ($n = 54,650$, excluding interchromosomal events) (Table 1) with MVNCall[25] using the Affymetrix 6.0 SNP chip based haplotype scaffolds employed for construction of the reference panel described previously[1].

**Table 1 | Characteristics of the consensus indel and structural variants set.**

| Type | Number | Genotyped | Validation rate (%) | Novel (%) | Rare (MAF <1%) | Low freq (1 <= MAF <5%) | Common (MAF >= 5%) | Mean Length | Length Stdev. | Load (avg. bp/ haplotype) | Load (avg. homozygous/ genome) |
|---|---|---|---|---|---|---|---|---|---|---|---|
| Indel | 1,739,300 | 1,739,300 | 98 | 11.5 | 46.5% | 15.5% | 38.1% | 2.5 | 2.8 | 633,765.5 | 363,862.0 |
| Complex indel | 52,913 | 52,913 | 80 | 82.8 | 25.7% | 17.5% | 56.8% | 9.6 | 9.0 | 123,765.9 | 72,082.9 |
| Deletion 21–100 bp | 24,167 | 22,914 | 99 | 21.6 | 21.5% | 14.6% | 63.9% | 35.9 | 17.4 | 230,838.6 | 160,802.6 |
| Deletion >100 bp | 19,840 | 17,636 | 93 | 41.9 | 49.2% | 13.3% | 37.5% | 3,908.2 | 21,507.3 | 3,099,740.4 | 1,928,806.6 |
| Mobile element insertion | 13,469 | 13,430 | 96 | 55.9 | 54.9% | 13.6% | 31.6% | n/a | n/a | n/a | n/a |
| Duplication | 1,738 | 601 | 85 | 83.9 | 82.5% | 8.2% | 9.3% | 61,322.2 | 947,459.0 | 482,059.4 | 169,250.2 |
| Inversion | 84 | 69 | 65 | 29.8 | 21.7% | 15.9% | 62.3% | 3,047,390 | 23,882,994.1 | 154,510.4 | 101,262.7 |
| Interchromosomal events | 60 | 46 | 83 | n/a | 39.1% | 10.9% | 50.0% | n/a | n/a | n/a | n/a |
| Novel segments | 7,718 | n/a | 90 | n/a | n/a | n/a | n/a | 561.3 | 2,000.8 | n/a | n/a |
| Large replacements: replaced segments | 281 | n/a | 98 | n/a | n/a | n/a | n/a | 6,053.5 | 27,970.3 | n/a | n/a |
| Large replacements: replacing segments | 281 | n/a | 98 | n/a | n/a | n/a | n/a | 1,272.4 | 2,018.9 | n/a | n/a |

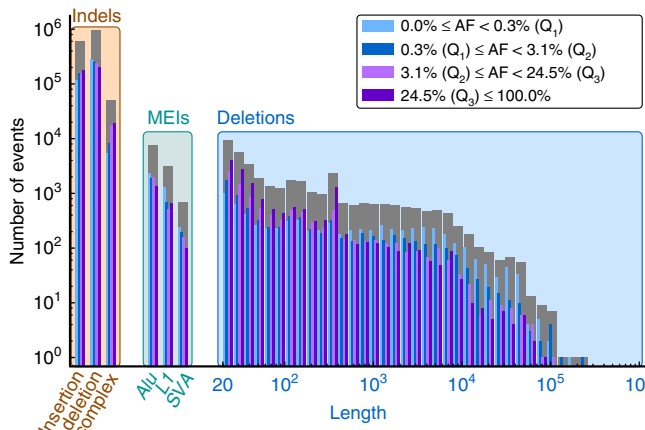

**Figure 2 | Number of simple and complex indels, mobile element insertions (MEIs) and deletions (stratified by length).** Grey bars correspond to total counts, whereas coloured (blue to violet) bars give counts stratified into four bins by allele frequency quartiles (Q1 to Q3).

**Linkage disequilibrium between tag SNPs and deletions.** To analyse the extent to which deletions are in linkage disequilibrium (LD; non-random association between alleles) with SNPs reported in the NHGRI Catalogue of genome-wide association studies[34] (GWAS), we tested all pairs of GWAS SNPs and deletions with a distance of at most 1 Mb ($n = 55,250$) for being in LD. Of these pairs, 14,003 (25.3%) showed statistically significant LD (based on Fisher's exact test, and controlling false discovery rate at 5% using with Benjamini-Hochberg's procedure, Supplementary Fig. 6). To assess whether this relatively high percentage of significant associations among the GWAS-SNP pairs is related to the GWAS status of the SNPs, we performed the same experiment on similar SNPs (applying a sampling technique previously described[35]) that were not associated through GWAS. We observed a significantly greater number GWAS SNP-deletion pairs (25.3%) in LD than of non-GWAS SNP-deletion pairs (19.1%, s.d. = 0.2), revealing that deletions deserve attention in studies of common genetic disorders and might be underlying some of the current GWAS SNP hits (Supplementary Figs 7 and 8). To test this hypothesis further, we filtered all GWAS SNP-deletion pairs for those in high LD ($r^2 \geq 0.8$), resulting in 115 pairs (Supplementary Data 6). Among these pairs, an exonic in-frame deletion (rs148114931) of 9 codons in *APOBR* appears twice, linking it to SNP rs151181 which has been associated to Crohn's disease[36] and SNP rs26528 associated to inflammatory

bowel disease[37]. Another deletion affected the UTR of *ITGA11,* which had been linked to major depressive disorder[38]. In addition, 61 intronic deletions were found to be in LD with SNPs previously associated to disease. In particular, due to the rigorous FDR correction applied, our catalogue of 115 significant GWAS SNP-deletion pairs provides strong initial evidence for further studies (Supplementary Data 6).

**Tag SNPs for SVs.** We considered SNPs represented on Affymetrix 6.0 array as well as known GWAS tagSNPs and compiled a list of SNP-SV pairs exhibiting a high degree of linkage disequilibrium ($r^2 \geq 0.8$ when considering 8,854 common, MAF >4%, deletions, 3,826 MEIs and 1,024 novel genomic segments). Beyond 115 deletions described above, other types of structural variants might be responsible for various human traits. Thus a total of 76 GWAS SNPs showed high LD ($r^2 \geq 0.8$) with at least one of 30 polymorphic MEIs or 43 new genomic segments (Supplementary Data 7). We expect that a significant part of these SVs might contribute to traits studied by GWAS studies.

**Imputing structural variants.** Genotype imputation, the prediction of missing genotypes based on a reference panel, has been very successful in boosting the power of GWAS, enabling meta-analyses across individual GWAS studies, and improving the chances of identifying causal variants by fine-mapping[39]. The value of the GoNL panel to robustly impute SNPs and indels has previously been shown[2]. We extend this concept and demonstrate that our SV-integrated panel allows for accurate imputation of SVs by imputing structural variants in an independent group of individuals based solely on their SNP genotype status (see Fig. 6 for a schematic overview). We genotyped all complex indels, deletions, duplications, inversions, and MEIs we found (Fig. 6, Step 1) in two independently sequenced Dutch genomes. We extracted all SVs that could be genotyped with a confidence of 0.999 (Step 2) in these individuals to create a set of gold standard genotypes. Gold standard genotype counts for each SV class are shown in Fig. 7a. SNP genotypes were filtered to only include those SNPs present on an Affymetrix 6.0 chip (Step 3), to simulate an array-based assay. On the basis of SNP genotypes obtained and the GoNL reference panel we used IMPUTE2 to impute SV genotypes (Step 4) to compare with the gold standard call. After imputation, SVs can optionally be filtered based on the genotype likelihoods (GLs). Here we document the performance for six different cutoffs (0.33, 0.75, 0.9, 0.95, 0.99 and 0.999). Step 5 thus only retains genotype calls that meet the respective threshold being tested. The quality of the remaining imputed genotypes was

determined by the fraction of imputed genotypes matching with the gold standard genotypes (concordance = squared correlation; see Supplementary Information, Section 3.6 for a formal definition). We refer to its inverse (that is, one minus concordance) as discordance. The GL threshold influences the tradeoff between discordance and the fraction of genotypes missing due to this filter (Fig. 7b). We find that more stringent filtering leads to an increase in concordance, demonstrating that the genotype likelihoods are meaningful. Specifically, employing the most stringent GL filter tested (at a level of 0.999) leads to only a moderate loss of 8.6–21.6% of genotypes imputed, depending on the variant type, while the discordance drops by 83% (from 2.9 to 0.5%) for duplications, by 71% (from 4.2 to 1.2%) for deletions, by 67% (from 6.9 to 2.3%) for MEIs, by 42% (from 14.2 to 8.2%) for inversions, and by 38% (from 10.5 to 6.5%) for complex indels. Based on Fig. 7b, we consider a threshold of 0.95 (red circle) a good tradeoff and report these results henceforth. With this setting, only 7.5% of all SVs are omitted, while the concordance is excellent; 88% for inversions, 92.4% for complex indels and 96.5% for MEIs, 98% for deletions and 99.1% for duplications. For the more common SV

classes with more than 1,000 genotype calls (Fig. 7a), namely complex indels, deletions, and MEIs, we further stratified performance based on genotypes (Supplementary Fig. 9) and on allele frequency. We observed that a higher MAF leads towards a higher discordance in genotypes across all three SV classes (Fig. 7c). This is driven by the fact that the vast majority of the rare variants in the GoNL reference panel are not found in these samples and is thus easy to impute correctly as homozygous reference. To investigate how well the rare allele could be imputed, we repeated the analysis restricted to only gold standard genotypes that contain one copy of the rare allele (that is, genotypes homozygous for the major allele are discarded), see Fig. 7d and Supplementary Fig. 10. Only the imputation of rare SV (or reference) alleles with panel frequencies below 5% leads to considerable losses in imputation quality, while imputation performance is excellent for higher MAFs (Supplementary Fig. 11). Finally, we compared the results of SV imputation using GoNL and 1,000 Genome Project[14] reference panels and a set of nearly 10,000 of structural variants detected in both projects. These results showed that SV imputation in two Dutch individuals using the GoNL panel is more sensitive and specific, particularly when imputing less frequent alleles (Supplementary Tables 4 and 5). The percentage of discordant calls was decreased when imputing with the GoNL panel from 2.7 to 1.2% for deletions called with a confidence of 0.95 and from 12.9 to 5.5% for minor allele deletions.

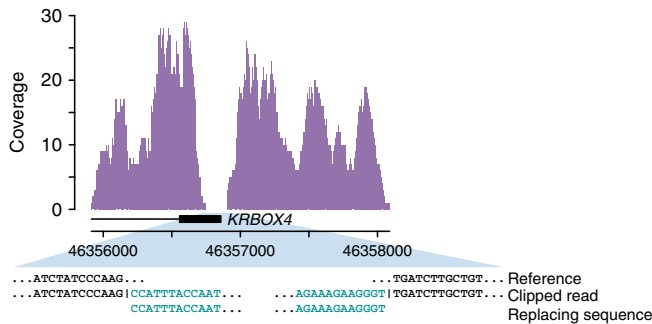

**Figure 3 | Example of a large replacement within the *KRBOX4* gene.** The plot depicts the coverage profile of whole genome sequencing reads from a GoNL sample with a homozygous replacement. The lack of coverage in the last exon of *KRBOX4* is coinciding with the position of the replacement. The breakpoint junctions of the replacement are indicated in the panel underneath the coverage plot.

## Discussion

The past few years have seen a remarkable progress in human genome sequencing studies, which has greatly improved our understanding of human genome variation[1,3,13,14,16–18,32]. These projects differ, often substantially, in terms of sample selection and sequence coverage. For example, to capture global diversity, the 1,000 Genomes Project selected 2,504 unrelated individuals from 26 populations and largely relied on the discoveries from low coverage whole genome data[13,14]. As a result a large proportion of common variants with population frequency greater than 1% have been discovered across multiple populations. In contrast, the UK10K project combined low coverage whole-genome with high coverage exome sequencing

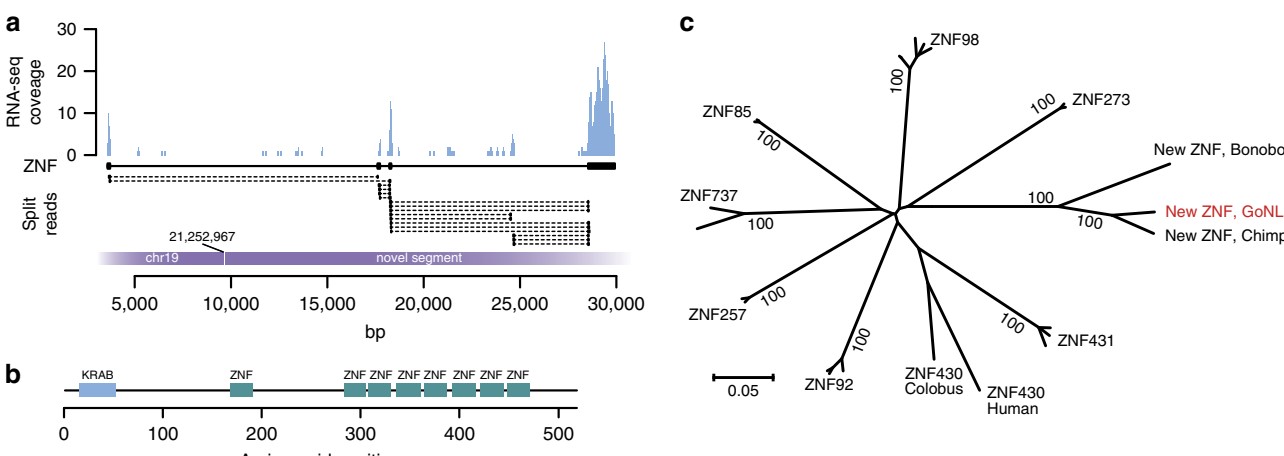

**Figure 4 | Identification and expression of a novel ZNF gene.** (**a**) A Geuvadis RNA-sequencing dataset (ERR188316) was mapped to the human reference genome, which was extended with a new genomic segment inserted in chr 19 (bp 21,252,967). The plot shows RNA expression and split-read mappings across the novel ZNF gene present on this new genomic segment. (**b**) Protein domain structure of the novel ZNF gene as determined using NCBI Conserved Domain Search. (**c**) Neighbor-joining tree built from alignment of protein sequences homologous to the novel ZNF gene. Values at the nodes indicate bootstrap support of each group. Distances indicate protein sequence divergence on amino acid level.

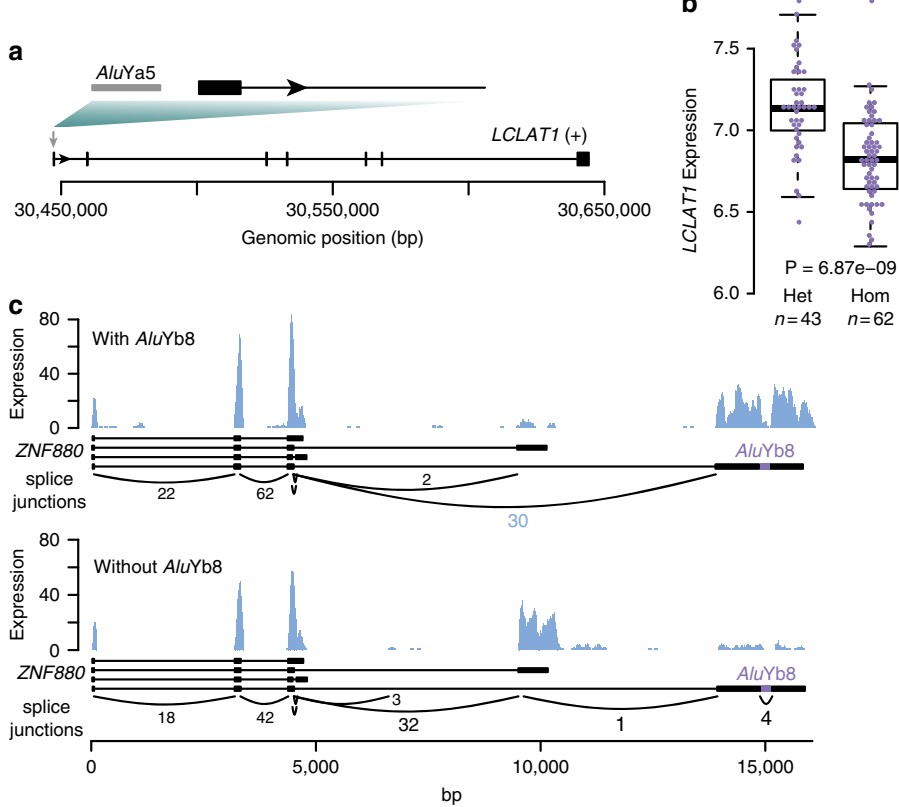

**Figure 5 | Effects of MEIs on gene expression.** (**a**) Schematic picture indicating an *Alu*Ya5 insertion in the promoter region of *LCLAT1*. (**b**) *LCLAT1* gene expression (log2 of normalized read count) in blood from GoNL individuals who are heterozygous (het) or homozygous (hom) for the *Alu*Ya5 insertion. (**c**) RNA expression effects of an *Alu*Yb8 insertion in the last exon of *ZNF880*. The presence of the *Alu*Yb8 element results in spliced transcripts, which preferentially contain the last exon, while the before last exon is skipped (upper panel). The reverse effect is seen in the absence of the *Alu*Yb8 insertion (lower panel).

approaches to identify rare variants associated with various genetic traits[17].

We exploit two features of the GoNL study design to create an SV-integrated reference panel. First, an elevated coverage allows for enhanced genotyping of SVs[18,22,23]. Second, the family-based design aids in establishing haplotypes across significantly longer ranges than achievable based on unrelated individuals[18,19]. Combining these two features yields a wealth of high-quality SV-integrated haplotypes, which we have corroborated by imputation experiments. In addition, the family design has facilitated analysis of variant transmission within a single generation. We have also been able to compile a list of SVs that are in high LD with disease associated SNPs which are highly unlikely to be false discoveries based on additional statistical analysis.

Our reference panel spans a wide range of variant classes, many of which have previously not been extensively reported, such as complex indels and medium-size SVs (affecting between 21 and 100 nucleotides). In particular medium-sized SVs are sometimes considered a blind spot in short read based variant discovery. This required method development for both discovery and genotyping, as well as clean sequencing library protocols. Furthermore we report a large collection of new genomic segments, representing several million bases missing from the genome reference.

Downstream analysis of the variants provided insights into the mutational dynamics, as well as the consequences of selection processes affecting structural variants. We show that the distribution and predicted functional impact of variants differs significantly between rare and common variants. While previous studies have demonstrated the effects of polymorphic deletions on gene expression[40], we here identified the effects on gene expression of additional forms of structural variation such as MEIs.

The evolution of high-throughput sequencing technologies, coupled with advances in data analysis, has leveraged substantial progress in variant detection in next-generation short-read data. Every new study has fostered our understanding about the human genome. Nonetheless, there is still considerable room for improvement[26]. Difficulties remain in capturing large and complex structural variants, especially those in repetitive regions. Evolving third-generation single molecule and long read sequencing, and further methodological advances such as global genome map technology, may further improve the discovery, genotyping and phasing of structural variants. Given present-generation technology, our approaches and the resulting reference panel provide both an advanced toolkit and a powerful resource, with great potential to decisively enhance genome-wide association and personalized genomics studies.

## Methods

**Sample collection and data generation.** Samples were collected as outlined in Boomsma *et al.*[19]. All participants provided written informed consent, and each biobank (LifeLines Cohort Study—University Medical Center Groningen, Leiden Longevity Study—Leiden Universitair Medisch Centrum, Netherlands Twin Registry—Vrij Universiteit Amsterdam, Rotterdam Study—Erasmus Medical Center, Rucphen Study—Erasmus Medical Center). Study protocol was approved by their respective institutional review board (IRB). The whole genome sequencing using the Illumina HiSeq 2000 platform using 90 bp paired-end reads[1]. Data were

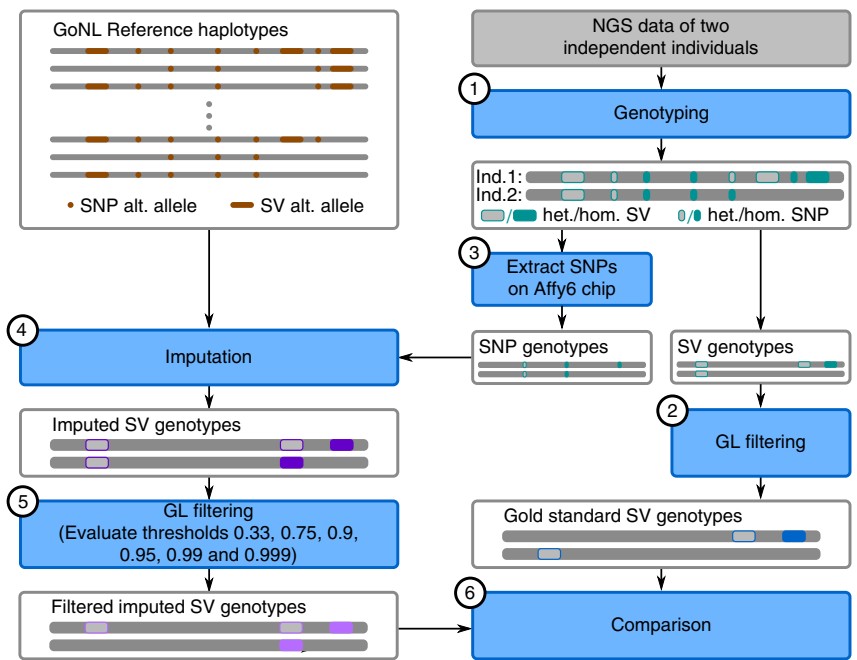

**Figure 6 | Schematic overview of the imputation experiment.** Haplotypes are represented by thin grey bars, whereas diploid chromosomes with genotype calls are indicated by thick grey bars. Processing steps are shown in blue, with numbers (in black circles) for being referenced in the main text.

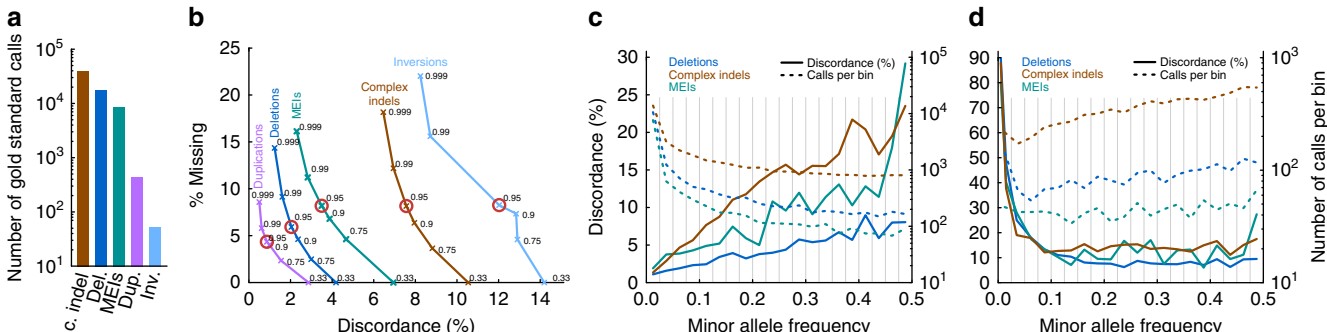

**Figure 7 | Imputation results for different SV types.** (**a**) Histogram on the number of gold standard genotype calls per SV class. (**b**) Relationship between discordance and fraction of missing genotypes when altering the genotype likelihood (GL) threshold used for filtering the imputed genotypes, ranging from 0.33 (no filter) to 0.999 across SV classes. Thresholds used for further analyses, including panels (**c,d**), are circled in red. Increasing the minimum GL results in fewer discordant genotypes but increases the number of missing genotypes. Imputation of inversions had the highest rate of discordance and missing genotypes, whereas the tandem duplications and deletions had lower rates of discordant and missing genotypes for those events with a high GL. (**c**) Discordance rates for deletions, complex indels and MEIs stratified by minor allele frequencies for 20 bins (width = 0.025). Bin boundaries are indicated by grey lines. The number of calls per bin are shown by dashed lines. (**d**) Same as (**c**), but restricted calls where the gold standard genotype contains at least one copy of the rare allele.

mapped to the UCSC human reference genome build 37 using BWA 0.5.9-r16 and quality control was performed as described earlier[1].

**Structural variation discovery.** We used 12 different algorithms for the discovery of structural variants, which use four different general approaches: split-read mapping, (discordant) read pairs, read depth, *de novo* assembly and combinations thereof, as shown in Fig. 1a. Details about how the individual methods were run are provided in Supplementary Information.

**Generation of consensus call set.** After creation of the algorithm specific calls sets a consensus set of indels and SVs were made for each to the SV types (indels, deletions, insertions, duplications, inversions, interchromosomal events and mobile element insertions). Events were merged per variant type using an algorithm-aware merging strategy (Supplementary Table 1). A consensus region was defined when overlapping regions were identified by 2 different detection strategies (for example split read and discordant read pair, see Supplementary Fig. 5 for the contributions of individual strategies for deletion detection, stratified by AF and event length),

and the boundaries of the event were determined by the algorithm with the highest breakpoint accuracy (as determined by the calling strategy) in combination with a 50% reciprocal overlap. For variants 20 bp and smaller in size an exact overlap was used, with support from at least two different methods.

**Validation experiments.** Validation was performed using PCR amplification of breakpoint junctions, and subsequent sequencing of the PCR products via Sanger or MiSeq sequencing. The validation set consisted of at least 96 candidates for indels, mid-size deletions, large deletions, MEIs, large replacements and novel segments, as well as 48 duplications, 76 inversions, 42 interchromosomal breakpoints and 10 complex indels (see Supplementary Data 1).

**Genotyping and phasing.** To genotype SVs, we used GATK's HaplotypeCaller for complex indels, MATE-CLEVER for deletions, Mobster for MEIs and Delly for inversions, duplications, and translocations. Details on how each tool was run are collected in Supplementary Information. For phasing, we used the haplotype scaffolds described earlier[1] to phase SVs onto the already phased sets of SNPs and

indels. The scaffold contains sites present on Affymetrix SNP 6.0 chips. Refer to the supplement of our previous study[1], Section 12, for details on how it was created. Phasing was done using MVNcall version 1.1 (ref. 25). We used the genotype likelihoods (GLs) reported by the genotyping tools described above. Before phasing, the GLs were regularized so as to avoid too low probabilities as detailed in the Supplementary Information.

**GWAS SNP permutation test.** For every GWAS SNP-deletion pair, we randomly selected a non-GWAS SNP-deletion pair that was similar in terms of potentially confounding variables, see Supplementary Information for those variables. We then applied Fisher's exact test and the Benjamini–Hochberg FDR control procedure on the matched non-GWAS SNP-deletion set and recorded the percentage of statistically significant pairs. This sampling procedure was repeated 1,000 times. The samples were found to have a mean of 19.1% and s.d. of 0.20, against the percentage of 25.3% for the GWAS SNPs (Supplementary Fig. 8), see Supplementary Information, Section 3.4, for further details.

**Imputation of structural variants.** SV genotyping of two independent Dutch individuals was done using the same pipeline as for genotyping SVs in the GoNL panel (see Supplementary Information), that is, GATK/HaplotypeCaller was used for complex indels, MATE-CLEVER for deletions, DELLY for duplications and inversions, and Mobster for MEIs. Genotype likelihoods (GLs) provided by these tools were used to determine the gold standard set, requiring a probability of 0.999 of the genotype being correct for all call types except for MEIs, where we used 0.85 account for differently calibrated GLs. For imputation of SV genotypes based on SNP genotypes and the GoNL panel, we used IMPUTE2 (ref. 41). Therefore, we first used SHAPEIT2 (ref. 42) for first phasing all SVs with the SNPs using the GoNL panel. Refer to Supplementary Information, Section 3.6 for the details such as command line arguments. Note that phasing genetic variants using SHAPEIT2 before imputing genotypes with IMPUTE2 follows best-practice recommendations (see IMPUTE2 https://mathgen.stats.ox.ac.uk/impute/impute_v2.html). The expected discordance between true and imputed genotype is based on comparing the probability distribution over the three different genotypes provided by IMPUTE2 for the imputed genotypes on the one hand, and the probability distribution provided by the read-based genotyping tools on the other hand, see Supplementary Information, Section 3.6 for details and definitions. We refer to the expected value of the discordance just as 'discordance' in the main text and we refer to 1-discordance as concordance.

**Data availability.** The references to depositories of software packages and parameters used in each analysis are given within corresponding sections of Supplementary Note. Sequence data have been deposited at the European Genome-phenome Archive (EGA), which is hosted by the European Bioinformatics Institute (EBI), under accession number EGAS00001000644, as well as genotyped variants discovered in this study, accession number EGAD00001002261. All variants calls, their allele frequencies and novel sequences are available from the official consortium webpage: http://www.nlgenome.nl. All other data are available from the corresponding authors on request.

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

## Acknowledgements

The GoNL Project is funded by the Biobanking and Biomolecular Research Infrastructure (BBMRI-NL), which is financed by the Netherlands Organization for Scientific Research (NWO project 184.021.007). BBMRI-NL funded K.Y. for validation experiments, CP2011-36. The Netherlands Organization for Scientific Research (NWO) has funded J.Y.H.-K. through Veni grant 016.166.015 and A.S. through Vidi grant 639.072.309. This work was supported, in part, by a U.S. National Institutes of Health (NIH) grant R01HG002385 and NIH U41HG007497 to E.E.E. E.E.E. is an investigator of the Howard Hughes Medical Institute.

## Author contributions

A.S., K.Y. and V.G. planned and directed the study. C.W., P.I.W.d.B., M.A.S., G.-J.B.v.O., P.E.S. and D.I.B. supervised GoNL data integration. J.Y.H.-K., T.M., W.P.K., L.C.F., J.A.B., L.J.D., A.A., V.K., D.T.T., R.W., B.P.C., P.D., J.dB, E.-W.L., F.v.D, F.H., A.S., K.Y. and V.G. performed data analysis. A.G.U., C.M.v.D, D.I.B., C.W. and P.E.S. provided samples for validation experiments. I.R., W.P.K. and V.G. performed the validation experiments and analysis. J.Y.H.-K., T.M., W.P.K., A.S., K.Y. and V.G. wrote the manuscript. E.E.E., C.W., P.I.W.d.B., M.A.S., G.-J.B.v.O., P.E.S., D.I.B. commented on the manuscript.

## Additional information

**Competing financial interests:** The authors declare no competing financial interests.

## The Genome of the Netherlands Consortium

Jasper A. Bovenberg[23], Anton J.M de Craen[19], Marian Beekman[19], Albert Hofman[17], Gonneke Willemsen[9], Bruce Wolffenbuttel[24], Mathieu Platteel[14], Yuanping Du[25], Ruoyan Chen[25], Hongzhi Cao[25], Rui Cao[25], Yushen Sun[25], Jeremy Sujie Cao[25], Pieter B.T Neerincx[14,16], Martijn Dijkstra[14,16], George Byelas[14,16], Alexandros Kanterakis[14,16], Jan Bot[26], Martijn Vermaat[15,27,28], Jeroen F.J Laros[15,27,28], Johan T. den Dunnen[15,27], Peter de Knijff[15], Lennart C. Karssen[17], Elisa M. van Leeuwen[17], Najaf Amin[17], Fernando Rivadeneira[10], Karol Estrada[10], Jouke-Jan Hottenga[9], V. Mathijs Kattenberg[9,28], David van Enckevort[28], Hailiang Mei[28], Mark Santcroos[29], Barbera D.C van Schaik[29], Robert E. Handsaker[6,30], Steven A. McCarroll[6,30], Arthur Ko[13], Peter Sudmant[13] & Isaac J. Nijman[4]

[23] Legal Pathways Institute for Health and Bio Law, Aerdenhout 2111XS, The Netherlands. [24] Department of Endocrinology University Medical Center Groningen, Groningen 9700RB, The Netherlands. [25] BGI, Shenzhen 518083, China. [26] Leiden Institute of Advanced Computer Science, Leiden University, Leiden 2300RC, The Netherlands. [27] Leiden Genome Technology Center, Leiden University Medical Center, Leiden 2300RC, The Netherlands. [28] Netherlands Bioinformatics Centre, Nijmegen 6245GA, The Netherlands. [29] Department of Clinical Epidemiology, Amsterdam Medical Center, Amsterdam 1090GE, The Netherlands. [30] Department of Genetics, Harvard Medical School, Boston 02115, USA

