## [Peer Review File · Nature Communications]

Reviewers' comments:

Reviewer #1 (Remarks to the Author):

The paper presents a haplotype reference panel that includes SNPs and SVs. This is an impressive piece of work. The analysis looks rigorous and is well presented. However, I think the authors could have done better at comparing to existing studies. I disagree with the sentence "current studies on SVs have failed to provide a global view of the full spectrum of SVs and to integrate them into reference panels of genetic variation." The 1000 Genomes Project did just this, and that should be acknowledged. Only the 2010 1000 Genomes paper is referenced, and the main project paper in Nature last year is not referenced. This is a major omission. Also, the author of ref4 is incorrect.

A comparison to the set of SVs found in the 1000 Genomes is needed. A table stratified by variant type, showing overlapping and novel variants is needed. Focus should be on the characteristics of the novel variants. Also, for overlapping SVs, is it better to impute from 1000 Genomes or from the GoNL data?

Reviewer #2 (Remarks to the Author):

NCOMMS-16-02152 "A high-quality reference panel reveals the complexity and distribution of structural genome changes in a human population"

Hehir-Kwa, Marschall, Klosterman et al. using 12 different tools and five algorithmic approaches computationally analysed whole-genome sequencing data (average coverage 14-15x reads) of 769 genomes from 250 parent-offspring trio families from the "Genome of the Netherlands" project (PMID: 23714750). They describe nine different classes of structural variants. Specifically, they demonstrate efficient detection of the very short (21-100 bp in size) CNVs that thus far have been under-recognized. Significant fractions of the reported variants (13.6% of indels and 38% of SVs) are novel. Over 800 identified variants were independently verified by Sanger sequencing or MiSeq. The Authors genotyped and long-range phased many of the identified variants and found linkage disequilibrium for several SNP-SV associations, further supported by imputation analyses. This significant progress in variant phasing is essential for interpretation of GWAS. They provide a list of 115 GWAS hitSNPs with significant LD with CNV deletions. In addition, they identified approx. 4.3 Mb of unannotated sequence with 11 novel potentially functional genes.

This work is a continuation and an important expansion of the analyses published in 2014 and 2015 (PMID: 24974849 and 25985141).

The manuscript is well written and is an important contribution to the literature.

I have only minor comments.

Page 6, Results

This yielded a confirmation rate for each variant class of between 80% to 97.9% with the exception of inversions (65.4%) which failed to produce a PCR product in 43.5% of the cases

Please clarify.

Page 7, Results

repetitive elements such as simple repeats (n=914) or segmental duplication (n=194).

Repetitive should read repeat.

Page 7, Results

by combining calls from discordant pair analysis with de novo genome assembly, we report 84 inversions and 60 of which 69 and 46 could be
This sentence is incomplete.

Page 8, top line

20bpt

Change to

20 bp

Page 10

and AluY insertions.

Do not italicize "Y".

Page 11

distance of at most 1MB

should read

distance of at most 1 Mb

Page 16

A consensus region was defined identified by 2 different detection strategies

Please revise this statement

Page 19

References 11 and 20 are incomplete.

Page 22

Heatmaps display the ALU insertions

should read

Heatmaps display the Alu insertions

Suppl Table 2

What is the difference between confirmed and verified?

There are multiple errors and typos.

Page 7, Results

Our data shows

Data are plural.

Suppl Page 7

We applied a GC correction and included CNVs in the the final call

Change to

We applied a GC correction and included CNVs in the final call

Suppl Page 24

standard deviation deviation

Suppl Page 26

For each familyy

Suppl Page 28
according

Suppl Page 28
genotyping

Page 39, Legend to Suppl Table 3
disequilibrium

Suppl Page 56
disequilibrium

REVIEWERS' COMMENTS:

Reviewer #1 (Remarks to the Author):

The authors have done a great job in responding to the comments.

This is an excellent paper and resource.

Reply to reviewers' comments:

Reviewer #1 (Remarks to the Author):

1.1 The paper presents a haplotype reference panel that includes SNPs and SVs. This is an impressive piece of work. The analysis looks rigorous and is well presented. However, I think the authors could have done better at comparing to existing studies. I disagree with the sentence "current studies on SVs have failed to provide a global view of the full spectrum of SVs and to integrate them into reference panels of genetic variation." The 1000 Genomes Project did just this, and that should be acknowledged. Only the 2010 1000 Genomes paper is referenced, and the main project paper in Nature last year is not referenced. This is a major omission. Also, the author of ref4 is incorrect.

We thank the reviewer for the positive comments on the scope and quality of our work. We agree that the results presented here should be more accurately placed within the context of previous studies. To this end variants presented here had been compared and annotated with data from a number of existing studies including the 1000 Genomes Phase 3 published last year (Sudmant et al, 2015 – ref 14, see Results, page 6), as well as other recent datasets (Sudmant et al, 2015 - ref 13; Chaisson et al, 2015 – ref 26).

We changed the statement in the abstract that now reads: "*However, the majority of studies provide neither a global view of the full spectrum of these variants nor integrate them into reference panels of genetic variation*".

We have also corrected reference 4.

1.2 A comparison to the set of SVs found in the 1000 Genomes is needed. A table stratified by variant type, showing overlapping and novel variants is needed. Focus should be on the characteristics of the novel variants. Also, for overlapping SVs, is it better to impute from 1000 Genomes or from the GoNL data?

Again, we thank the reviewer for the useful suggestions. We updated table 1 that now includes percentage of novel variants (as compared to dbSNP, CHM1 and 1000 genome project data) and stratified by variant type. We have also performed the imputation of SV set that is common between GoNL and 1000 genomes phase 3 datasets using either GoNL or the 1000 Genomes panels. These results are presented in the Results section: Imputing structural variants, page 13 and include new Supplementary Tables 11 and 12. In particular, we observed an improvement when imputing the alleles that are less frequent in a population. We appended a paragraph to the results section that reads: "*Finally, we compared the results of SV imputation using GoNL and 1000 genome project¹⁴ reference panels and a set of nearly 10 thousand of structural variants detected in both projects. These results showed that SV imputation in two Dutch individuals using the GoNL panel is more sensitive and specific, particularly when imputing less frequent alleles (Supplementary Tables 11 and 12). The percentage of discordant calls was decreased when imputing with the GoNL panel from 2.7% to 1.2% for deletions called with a confidence of 0.95 and from 12.9% to 5.5% for minor allele deletions*".

Reviewer #2 (Remarks to the Author):

I have only minor comments.

2.1 Page 6, Results

This yielded a confirmation rate for each variant class of between 80% to 97.9% with the exception of inversions (65.4%) which failed to produce a PCR product in 43.5% of the cases

Please clarify.

We rephrased and extended the part on validation results that should become more detailed and clear and currently reads: *“This yielded a confirmation rate for each variant class of between 80% to 98.6% with the exception of inversions (64.5%), which failed to produce a PCR product in 35.5% of the cases (Table 1). The high rate of PCR failures for inversions might either be a result of false positives, or stem from poor performance of PCR assays, given the frequent occurrence of repetitive and complex sequences at inversion breakpoints”*.

2.2 Page 7, Results

repetitive elements such as simple repeats (n=914) or segmental duplication (n=194).

Repetitive should read repeat.

Corrected

2.3 Page 7, Results

by combining calls from discordant pair analysis with de novo genome assembly, we report 84 inversions and 60 and of which 69 and 46 could be

This sentence is incomplete.

We have updated this sentence; it now reads: *“...of which 69 and 46 could be successfully genotyped”*.

2.4 Page 8, top line

20bpt

Change to

20 bp

Corrected

2.5 Page 10

and AluY insertions.

Do not italicize "Y".

Corrected

2.6 Page 11

distance of at most 1MB

should read

distance of at most 1 Mb

Corrected

2.7 Page 16

A consensus region was defined identified by 2 different detection strategies

Please revise this statement

We have updated this sentence, it now reads: *“A consensus region was defined when overlapping regions were identified by 2 different detection strategies (for example split read and discordant read pair, see Supplementary Fig. 5 for the contributions of individual strategies for deletion detection, stratified by AF and event length), and the boundaries of the event were determined by the algorithm with the highest breakpoint accuracy (as determined by the calling strategy) in combination with a 50% reciprocal overlap. For variants 20bp and smaller in size an exact overlap was used, with support from at least two different methods”*.

2.8 Page 19

References 11 and 20 are incomplete.

We have corrected these references.

2.9 Page 22

Heatmaps display the ALU insertions
should read

Heatmaps display the Alu insertions

Corrected, now reads: "Heatmaps display the insertions of *Alu*, L1 and SVA elements".

2.10 Suppl Table 2

What is the difference between confirmed and verified?

This inconsistency in the Supplementary Table 2 has been corrected (all occurrences of 'confirmed' replaced by 'verified')

There are multiple errors and typos.

Page 7, Results

Our data shows

Data are plural.

Corrected

Suppl Page 7

We applied a GC correction and included CNVs in the the final call

Change to

We applied a GC correction and included CNVs in the final call

Corrected

Suppl Page 24

standard deviation deviation

Corrected

Suppl Page 26

For each familiy

Corrected

Suppl Page 28

according

Corrected

Suppl Page 28

genotyping

Corrected

Page 39, Legend to Suppl Table 3

disequalbrium

Suppl Page 56
disequilibrium
Corrected